# Effective Degradation of Rh 6G Using Montmorillonite-Supported Nano Zero-Valent Iron under Microwave Treatment

**DOI:** 10.3390/ma11112212

**Published:** 2018-11-07

**Authors:** Wenxiu Rao, Hao Liu, Guocheng Lv, Danyu Wang, Libing Liao

**Affiliations:** 1Beijing Key Laboratory of Materials Utilization of Nonmetallic Minerals and Solid Wastes, National Laboratory of Mineral Materials, School of Materials Science and Technology, China University of Geosciences, Beijing 100083, China; 2103170029@cugb.edu.cn (W.R.); 13020023660@163.com (D.W.); 2School of Science, China University of Geosciences, Beijing 100083, China

**Keywords:** nano zero-valent iron, montmorillonite, organic dyes, Fenton-like reaction, microwave

## Abstract

Nano zero-valent iron has drawn great attention for the degradation of organic dyes due to its high reactivity, large specific surface area, lightweight, and magnetism. However, the aggregation and passivation of iron nanoparticles may prohibit the wide use of it. A new composite material was prepared by loading nano zero-valent iron (nZVI) on montmorillonite (MMT) to overcome the above shortcomings and it was further used for the removal of Rhodamine 6G (Rh 6G) under microwave treatment in the present work. The effects of various parameters, including the initial concentration of Rh 6G, microwave power, and pH value were investigated. The new composite material (nZVI/MMT) showed an excellent degradation ability for removing Rh 6G, and the removal amount reached 500 mg/g within 15 min. The degradation rate reached 0.4365 min^−1^, significantly higher than most previous reports using other removal methods for Rh 6G.

## 1. Introduction

Printing and dyeing wastewaters are the major sources for water pollution and elsewhere [1]. Every year, a large amount of these wastewaters with high levels of organic pollutants and complex components are discharged [2], which will greatly affect the water security of living creatures [3]. In addition, large amounts of pollutants are produced during the production of organic dyes in the textile industry. Serious environmental pollution will occur without treatment [4]. However, they are difficult to be naturally biodegraded [5]. Therefore, seeking efficient and economical treatment methods for dye wastewater are urgently needed in the dye production and printing/dyeing industries.

Many dye wastewater treatment technologies have been explored [6], including physical methods [7], chemical methods [8], biological methods [9], and so on. Of these methods, the Fenton method is an effective method for dye degradation [10]. The traditional Fenton method utilizes Fe^2+^ to catalyze H_2_O_2_ to produce hydroxyl radicals (•OH) with an oxidation-reduction potential of 2.80 V [11]. The hydroxyl radical can directly degrade most of the organic pollutants [12]. Nevertheless, the traditional Fenton method requires a concentration of iron ions of 50–80 mg/L [13], which is much higher than the allowance of 2 mg/L of the EU Steering Committee’s requirement. In addition, a large amount of iron sludge will be produced during the reaction [14], and the optimum pH value for the traditional Fenton method is 3 [15]. Thus, the pH value of wastewater needs to be adjusted, increasing the operating costs of water treatment.

Fenton-like reactions can occur under neutral conditions [16]. The Fenton reaction nano zero-valent iron (nZVI) has degraded many organic substances [17,18]. In recent years, nZVI has attracted much attention due to the high specific surface area, large surface energy, and excellent performance for treating contaminants [19]. The nZVI had good results in the treatment of printing and dyeing wastewater [20], heavy metals [21,22], chlorinated organic compounds [23,24], organic insecticides [25,26], and other areas [27,28]. However, nZVI is highly reactive and easily oxidized [29], limiting its practical applications. Montmorillonite (MMT) is a 2:1 type of mineral, which consists of two sheets of silicon tetrahedron and a middle sheet of aluminum octahedron [30]. Because of its unique layered structure, MMT can adsorb a variety of organic contaminants, such as chlorpheniramine [31], benzothiazolium [32], and Rh 6G [33]. In addition, the cations are stably arranged in the interlaminar domain of the MMT lamellae due to a strong electrostatic attraction force between the cations and interlaminar layer [34]. Because more and more composites are used as catalysts [35,36], we expect that the MMT can protect the nZVI to avoid its oxidation by loading nZVI into MMT lamellae to form the composite.

Using light [37], ultrasound [38], or microwave [39] can speed up the Fenton reaction speed. In these methods, because of the strong penetrating effect of the microwave, it can directly heat the reactant molecular and provide more energy to the reaction system [40]. During these interactions, the activation energy of the reaction and the chemical bond strength of the molecular can be reduced, while the production rates of Fe^2+^ and •OH speed up, leading to the acceleration of the rate of the degradation reaction [41].

In the present work, nZVI/MMT was synthesized using a liquid phase reducing method and used as a degradable material to deal with Rh 6G under microwave treatment. The factors that affect the reaction efficiency and rate were discussed in detail. The removal amount of Rh 6G reached 500 mg/g within 15 min and its degradation rate reached 0.4365 min^−1^, significantly higher than previous reports using other removal methods for Rh 6G. This work provides a good reference for the effective treatment of printing and dyeing wastewater using nZVI materials.

## 2. Materials and Methods

### 2.1. Materials

Iron(III) chloride hexahydrate (FeCl_3_·6H_2_O), sodium borohydride (NaBH_4_), hydrochloric acid (HCl), sodium hydroxide (NaOH), indolepropionic acid (IPA), p-benzoquinone (PBQ), triethanolamine (TEOA), and silver nitrate (AgNO_3_) were bought from Beijing Chemical Workstation (Beijing, China). The Rh 6G was from Aladdin Holdings Group (Beijing, China). The MMT was obtained from Inner Mongolia Ningcheng Tianyu Bentonite Technology Co., Ltd. (Ningcheng, China), and used without further purification. The chemicals were all of the analytical grade. All of the solutions were formulated with distilled water.

### 2.2. Synthesis of nZVI/MMT

In a three-necked flask, 10 g of FeCl_3_·6H_2_O was dissolved in the solvent containing 30 mL of deionized water and 20 mL of ethanol. Then 5 g of MMT was added to the solution. NaBH_4_ solution (8 g dissolved in 40 mL distilled water) was added dropwise into the mixture that had been high speed stirred for 4 h and blown with high purity nitrogen for 30 min. After the addition of NaBH_4_, the mixture was stirred for another 40 min at room temperature in nitrogen. The solid part was freeze-dried after centrifugation. 

### 2.3. Characterizations

The structures of the samples were determined by the X-ray diffractometer (XRD) with a CuKα-radiation at 30 kV and 20 mA (Rigaku D/Max-IIIa X-ray diffractometer, Rigaku, Tokyo, Japan). Samples were scanned from 3° to 70° at a speed of 8° min^−1^ with a scanning step length of 0.01°. TEM (transmission electron microscopy) tests of the MMT and nZVI/MMT were carried on a JEOL-2100 transmission electron microscope (JEOL, Tokyo, Japan) with an accelerating voltage of 200 kV. The sample powders were well dispersed in ethanol and dried on a copper grid supported carbon film before observation. Microwave absorption properties of the flexible absorbing film were analyzed using a microwave network analyzer N5244A (Agilent Technologies, Santa Clara, CA, USA). The frequency range was from 2 to 10 GHz. The coaxial wire method was adopted for the analyses.

### 2.4. Rh 6G Removal Experiment

Rh 6G was dissolved in distilled water to prepare the Rh 6G solution (5000 mg/L) and left to rest overnight. Zero point one grams of nZVI/MMT was added to 10 mL of Rh 6G stock solution in a glass beaker and stirred in batch experiments. Then the microwave reactor was used to treat the mixture under different microwave powers. After microwaving, the cooled slurry was centrifuged at 6000 rpm for 5 min. Then 10 mL aliquot of the supernatant was filtered with a syringe filter and transferred to a 50 mL colorimetric tube for UV-visible analysis at the end of the process. In the cycle test, nZVI/MMT was prepared using an NaBH_4_ reduction of the Fe^3+^ back to Fe^0^. All experiments were run in duplicates.

## 3. Results and Discussion

### 3.1. Crystallization, Morphology and Valence States of nZVI/MMT

We developed a new composite material for Rh 6G degradation under microwave treatment, and the composite material was synthesized using a liquid phase reducing method (Section 2.2) to load nZVI on MMT (Figure 1a), which greatly improved the antioxidant and anti-agglomerating ability of nZVI. The phase structures of the as-prepared MMT, nZVI, nZVI/MMT were examined by XRD (Figure 1b). The 2θ values in the XRD pattern of the nZVI/MMT locate at 5.89° and 19.71°, corresponding to 001 and 100 basal planes of MMT (JCPDS No. 47-1097). The reflection at 44.7° is owed to the 110 basal planes of Fe^0^ (JCPDS No. 06-0696). No iron oxide peak was found in the nZVI/MMT, indicating that the iron was not oxidized during the synthesis process, or only a very small portion of amorphous iron oxides formed. In addition, the pattern of nZVI/MMT is similar to that of pure MMT, suggesting that the structure of the nZVI/MMT remains before and after the loading of nZVI.

In order to further reveal the morphologies of the samples, TEM was carried out (Figure 1). Compared with pristine MMT (Figure 1c), lots of zero-valent iron nanoparticles with an average size of 15 nm are uniformly dispersed on the surface of nZVI/MMT (Figure 1d). This indicates that MMT could be used as a support to prevent the aggregation of nZVI. The energy dispersive spectrum analysis reveals that this sample composes of Fe, Si, Al, Na elements and no impurities are found (Figure 1e).

The antioxidant ability of nZVI/MMT and nZVI was compared by observing their color change at different time slots in the air, as shown in Figure 2. Both fresh solutions are black. As time increases, due to the oxidation of nZVI, both solutions’ color gradually become lighter. Nevertheless, the nZVI/MMT shows much better antioxidant ability than nZVI. For example, the black nZVI solution totally turns to yellow at 72 h. In contrast, the nZVI/MMT solution just begins to turn to yellow at the same time, indicating that only a small portion of nZVI is oxidized at this moment. The above results prove that the zero valent iron loaded on the MMT is more difficult to be oxidized than the nZVI alone, which brings a big advantage for the application of nZVI/MMT.

The compositions and the valence of the iron on the surface of nZVI/MMT were analyzed using XPS technique. Figure 2e shows the survey spectrum of nZVI/MMT, revealing that the surface is composed of iron elements. Figure 2f shows the fine spectrum of Fe 2p of nZVI/MMT comprises four different peaks. The iron content of the composite material is 15.35%, higher than pure MMT (2.01%). The peaks at binding energies of 706.3 eV and 719.7 eV could be owed to the 2p3/2 and 2p1/2 peaks of Fe^0^, respectively [42]. The other two peaks at binding energies of 711.5 eV and 725.0 eV could be attributed to the 2p3/2 and 2p1/2 peaks of Fe(III), respectively, revealing that partial Fe^0^ on the surface had been oxidized to Fe(III).

### 3.2. Microwave Absorption Properties of nZVI/MMT

Microwave treatment can cause heterogeneous localized heating, called “hot spots”, which may significantly promote the rate of free radical generation [43]. The microwave network analysis characterizes the microwave absorption characteristics of nZVI/MMT nanocomposites. Generally speaking, the microwave absorbing performance of nZVI/MMT was characterized by the reflection loss (RL), which could also be determined by the complex permittivity and permeability for the given frequency and absorber thicknesses. The effective interaction between the complex permittivity and the permeability of nZVI/MMT could produce significant RL. The RL of the as-prepared composites could be calculated using the following equation [44]:(1)$$\mathrm{RL}(\mathrm{dB})=20\log\left|{(Z}_{\mathrm{in}}-1)/{(Z}_{\mathrm{in}}+1)\right|$$
(2)$$Z_{\mathrm{in}}={\left(\mu_{\gamma }/\varepsilon_{\gamma }\right)}^{1/2}\tanh\left[j(2\pi \pi f/c){\left(\mu_{\gamma }\varepsilon_{\gamma }\right)}^{1/2}\right]$$
where Z_in_ represents the normalized input impedance relating to the impedance in free space, f represents the frequency of microwave absorbing, d represents the thickness of the absorbing layer, c represents the velocity of electromagnetic wave in vacuum, and ε_r_ and u_r_ represent the relative complex permittivity (ε_r_ = ε’ − jε”) and permeability (µ_r_ = µ’ − jµ”), respectively.

The microwave absorption of the dielectric loss and magnetic loss are two factors that affect RL. The ability to store electric and magnetic energy is a reaction to the real permittivity (ε’) and the real permeability (µ’), while the dissipation of electric and magnetic energy is related to the imaginary permittivity (ε”) and the imaginary permeability (µ”) [45]. The dielectric loss (tan δe = ε”/ε’) for nZVI/MMT gradually increases while magnetic loss (tan δμ = μ”/μ’) decreases gradually from 2 to 5 GHz (Figure 3a). Both tan δe and tan δμ stay at a relatively fixed value from 5 to 10 GHz. Therefore, dielectric loss plays a major role in controlling microwave absorption, and magnetic loss plays a minor role. Moreover, the value of the dielectric loss is higher than the magnetic loss, suggesting that dielectric loss contributes more to enhancing the microwave absorption properties of nZVI/MMT, rather than the magnetic loss.

The RL values of nZVI and MMT are very close to 0 from 2 to 10 GHz, while the RL value of nZVI/MMT shows a strong absorbing peak at the frequency of 5.3 GHz (Figure 3b). This means that nZVI/MMT had a much better microwave absorbing ability and could be used as an excellent microwave absorption material due to its high bandwidth and broad spectrum characteristics.

### 3.3. Degradation of Rh 6G Using nZVI/MMT

Nano zero-valent iron can degrade many organic dyes using the following Fenton-like Reactions (3)–(5) [46,47] because the produced •OH and •O_2_^−^ have a strong degradation ability for a great majority of organic pollutants. Furthermore, the generated rate of the above free radicals can be improved under microwave treatment.

(3)$$O_{2}+{\mathrm{Fe}}^{0}+{2H}^{+}\overset{}{\to }{\mathrm{Fe}}^{2+}+H_{2}O_{2}$$

(4)$${\mathrm{Fe}}^{2+}+H_{2}O_{2}\overset{}{\to }{\mathrm{Fe}}^{3+}+\cdot \mathrm{OH}+{\mathrm{OH}}^{-}$$

(5)$${\mathrm{Fe}}^{2+}+O_{2}\overset{}{\to }\cdot O_{2}{}^{-}+{\mathrm{Fe}}^{3+}$$

The degradation ability for dyes of nZVI/MMT under microwave treatment was evaluated by removal of Rh 6G. We studied the effect of different initial concentrations of Rh 6G on the removal efficiency (Figure 4a). When the initial concentration of Rh 6G is less than 5000 mg/L, the maximal removal amount of Rh 6G increases with the increase of the initial concentration of Rh 6G; when the initial concentration is larger than 5000 mg/L, the maximal removal amount maintains at 500 mg/g. Therefore, an optimal initial concentration of Rh 6G was set as 5000 mg/L in the rest of the experiments.

The effect of microwave power on the removal amount of Rh 6G by nZVI/MMT was investigated (Figure 4b). The degradation rate of Rh 6G gradually increases with the increase of microwave power as the increasing microwave power will accelerate the generation of free radicals. When the microwave power is 700 W, the removal amount of Rh 6G reaches 500 mg/g. We further performed experiments to investigate the effect of pH conditions on the removal amount of Rh 6G using nZVI/MMT (Figure 4c). Because the Fenton reagents are strongly affected by pH, the degradation ability under acidic conditions is significantly better than that under alkaline conditions. When the pH is high, the generation of hydroxyl radicals is inhibited and the iron ions are precipitated as hydroxides, leading to a decline in degradation efficiency. In addition, H_2_O_2_ would be decomposed ineffectively under alkaline conditions.

The degradation abilities of Rh 6G (initial concentration of 5000 mg/L) using nZVI, MMT, and nZVI/MMT composite were compared under a microwave power of 700 W (Figure 4d). The removal concentration of Rh 6G within 15 min using nZVI/MMT composite is 500 mg/g, while it is 471 mg/g and 483 mg/g using MMT and nZVI, respectively, better than MMT on Rh 6G adsorption and nZVI degradation. The enhanced performance of nZVI/MMT composite mainly results from the uniform dispersion of the nZVI on the MMT, which prevents the oxidation of the nZVI. Therefore, MMT loaded nZVI composite is an effective treatment material for organic wastewater. The natural logarithm Ln (C_0_/C) of Rh 6G at time (t) to the initial Rh 6G was plotted and linearly fitted (Figure 4e), which reveals that it is the first order kinetics of oxidative degradation with a linear correlation coefficient of 0.9976 and the apparent reaction rate constant is 0.4365 min^−1^. As summarized in Figure 4f, this reaction rate is significantly higher than previous reports using other removal methods for Rh 6G [48,49,50,51,52].

The concentration of Rh 6G solution without nZVI/MMT did not change with the increase of reaction time under microwave treatment (Figure 5a). In contrast, the concentration of Rh 6G decreased significantly when nZVI/MMT was added (Figure 5b), and Rh 6G can be completely degraded at around 15 min.

The used composite material can be regenerated using NaBH_4_ to reduce Fe^3+^ as Fe^0^. Figure 5c shows the cyclability of the regenerated nZVI/MMT composite, and its removal rate can still remain more than 90% after five times of recycling, indicating that the prepared material can be reused several times.

As mentioned before, nZVI can activate the molecular oxygen in the solution under the treatment of microwaves to produce strong oxidizing radicals of •O_2_^−^ and •OH, and these active substances can lead to efficient oxidative degradation of organics. In order to study the role of these free radicals, same molar concentrations of indolepropionic acid (IPA), p-benzoquinone (PBQ), triethanolamine (TEOA), and silver nitrate (AgNO_3_) were added to the Rh 6G solution during degradation. TEOA is a typical hole-trapping agent [53] and AgNO_3_ is a typical electron capturing agent [54]. Figure 5d displays the degradation percentages of Rh 6G with different additives, revealing that the degradation ability of Rh 6G gets worse with adding IPA or PQB, indicating that holes are not a key component in the degradation of Rh 6G, and electrons have a weak effect on the degradation of Rh 6G. Therefore, the number of •OH or •O_2_^−^ radicals can greatly reduce in the Rh 6G degradation process after adding IPA or PQB scavengers, leading to the decrease of degradation ability of Rh 6G. The above results also confirm the essential role of •OH or •O_2_^−^ in the degradation of Rh 6G using nZVI/MMT under microwave treatment.

## 4. Conclusions

In summary, a nZVI/MMT composite was synthesized by loading nZVI on MMT, showing better antioxidation ability than pure nZVI and strong microwave absorbing ability. The nZVI/MMT composite was further used to degrade Rh 6G under microwave treatment. As compared to nZVI and MMT, nZVI/MMT showed an excellent degradation ability for the removing of Rh 6G. For example, the corresponding removal amount reached 500 mg/g within 15 min and its degradation rate reached 0.4365 min^−1^, significantly higher than previous reports using other removal methods for Rh 6G. In addition, nZVI/MMT can be used several times with a slight decrease in the removal amount. With the usage of different scavengers, it was confirmed that the •O_2_^−^ and •OH radicals were formed and played an important role during the Rh 6G degradation. The proposed nZVI/MMT composite in the present work demonstrated simplicity, flexibility, short reaction times, and high removal efficiency for organic dyes. Therefore, the prepared composite material has a good application prospect in wastewater treatment.

## Figures and Tables

**Figure 1 materials-11-02212-f001:**
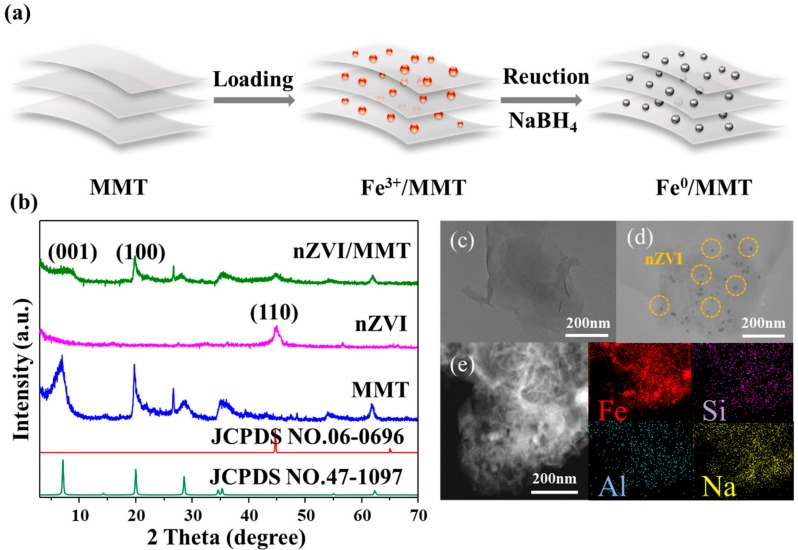
(**a**) Schematic for the synthesis of the nano zero-valent iron/montmorillonite (nZVI/MMT) composite; (**b**) X-ray diffraction patterns of MMT, nZVI and nZVI/MMT; (**c**) TEM image of MMT; (**d**) nZVI/MMT; (**e**) Energy dispersive spectrometer (EDS) element mapping image of nZVI/MMT.

**Figure 2 materials-11-02212-f002:**
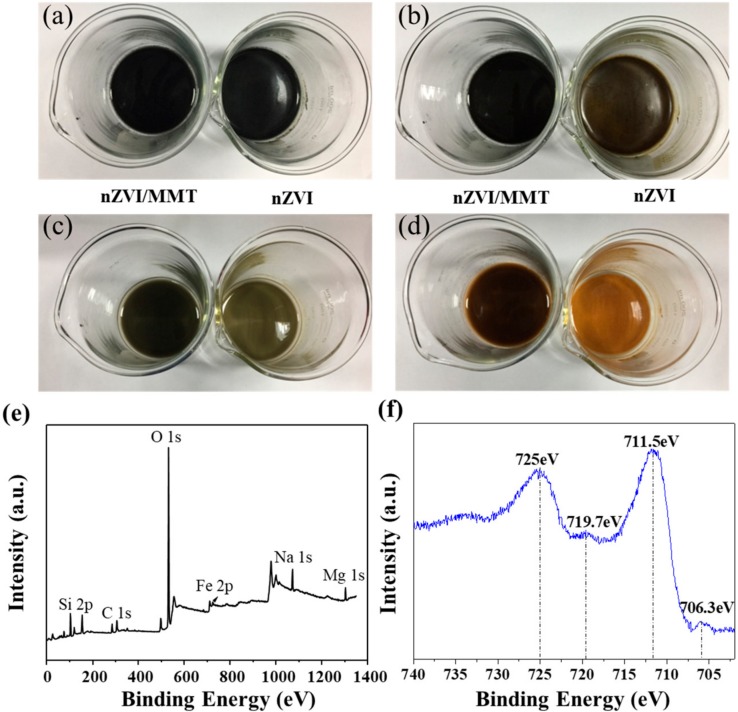
Color changes of nZVI and nZVI/MMT in water taken around: 0 h (**a**), 12 h (**b**), 36 h (**c**), and 72 h (**d**); X-ray photoelectron spectroscopy (XPS) survey spectrum of nZVI/MMT (**e**); and XPS fine spectrum of Fe 2p (**f**).

**Figure 3 materials-11-02212-f003:**
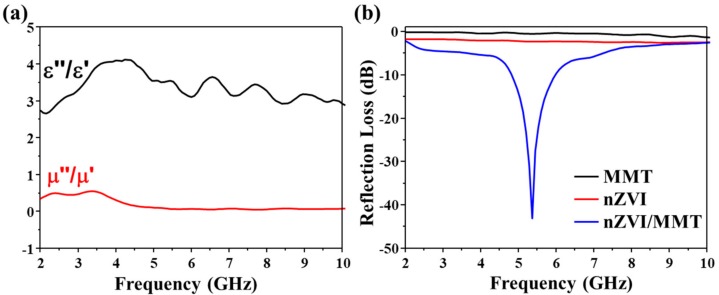
Relative complex permittivity and relative complex permeability’s frequency dependence of nZVI/MMT (**a**), and reflection-absorption rate of MMT, nZVI and nZVI/MMT (**b**).

**Figure 4 materials-11-02212-f004:**
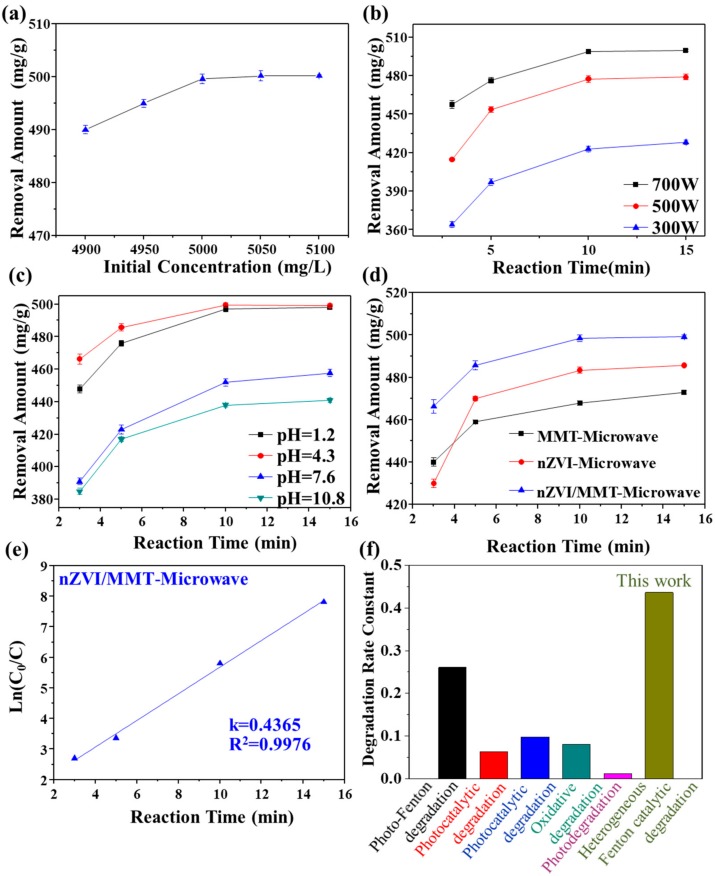
Removal amount of Rh 6G with the different initial concentration of Rh 6G (**a**), microwave power (**b**), solution pH (**c**); removal amount of Rh 6G in the presence of MMT, nZVI and nZVI/MMT under microwave treatment (**d**); pseudo-first-order reaction model for Rh 6G removal by nZVI/MMT (**e**); and comparison of the reaction rates of different treating methods for removing Rh 6G (**f**)**.**

**Figure 5 materials-11-02212-f005:**
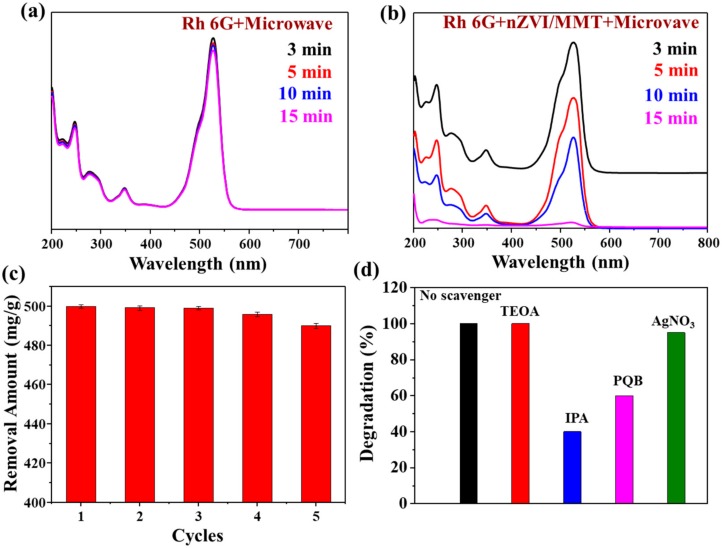
UV/Vis spectra of Rh 6G solution without nZVI/MMT after microwave for different time (**a**), with nZVI/MMT after microwave for different time (**b**), cycle stability test for nZVI/MMT (**c**), and degradation percentages of Rh 6G with different additives using nZVI/MMT under microwave treatment (**d**). Triethanolamine (TEOA), indolepropionic acid (IPA), p-benzoquinone (PBQ), and silver nitrate (AgNO_3_).

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
