# Peer review of "Effective Degradation of Rh 6G Using Montmorillonite-Supported Nano Zero-Valent Iron under Microwave Treatment"

_materials, 2018, doi:10.3390/ma11112212_

Round 1
Reviewer 1 Report
The paper presents the synthesis and characterization of montmorillonite-supported iron nanoparticles used for the degradation of organic dyes.
The layout of the article is correct. Both the introduction and the remaining chapters contain the necessary information, but I do have several concerns and questions:
1. Keywords: Add “organic dyes” to keywords.
2. There are double spaces between words (units) throughout the text.
3. Chapter 2.4: The concentration of RH 6G in the model solution is very high. Please provide some information on the actual concentrations of dyes in wastewater.
4. Fig. 4. describes results conducted on water solutions with different concentrations of RH 6G and different pH. Add this information to the Materials and Methods chapter. What reagents were used to correct pH?
5. Chapter 2.4, line 93: What microwave powers were used?
6. Chapter 3 contains a lot of information about several organic solvents and other chemical reagents (line 218,219), which were used in the experiment. They should be listed in Chapter 2.1.
7. Chapter 3.2., line 144: ... by equation (1) and (2):........
Were these equations formulated by the authors? Give a reference to them.
8. There are no references to reactions (3), (4) and (5) in the text. I guess that they are Fenton-like reactions.
9. The first sentence of Chapter 3.3. should be written as follows: “INP can degrade many organic dyes through the following Fenton-like reactions (3) – (5) [45,46]...”
10. Fig. 4.: What method was used to calculate the error bars presented in the figure?
11. Fig. 5: The OX axis of Fig. 5d. should be described.
Author Response
The paper presents the synthesis and characterization of montmorillonite-supported iron nanoparticles used for the degradation of organic dyes.
The layout of the article is correct. Both the introduction and the remaining chapters contain the necessary information, but I do have several concerns and questions:
1. Keywords: Add “organic dyes” to keywords.
We have added “organic dyes” to keywords.
2. There are double spaces between words (units) throughout the text.
We have changed it in the text.
3. Chapter 2.4: The concentration of RH 6G in the model solution is very high. Please provide some information on the actual concentrations of dyes in wastewater.
Thank you very much for your suggestion, the actual concentrations of dyes in wastewater can not reach such high 5000mg/L, the removal amount can still reach 500mg/g When the concentration of Rh 6G is 5000 mg/L, indicating the composite has good removal performance for Rh 6G.
4. Fig. 4. describes results conducted on water solutions with different concentrations of RH 6G and different pH. Add this information to the Materials and Methods chapter. What reagents were used to correct pH?
We have added reagents in the Materials and Methods chapter.
5. Chapter 2.4, line 93: What microwave powers were used?
The microwave frequency of microwave network analyzer is 2-10 GHz, and microwave powers are 300-700 W when reaction proceeds.
6. Chapter 3 contains a lot of information about several organic solvents and other chemical reagents (line 218,219), which were used in the experiment. They should be listed in Chapter 2.1.
We have added reagents in the Materials and Methods chapter.
7. Chapter 3.2., line 144: ... by equation (1) and (2):........Were these equations formulated by the authors? Give a reference to them.
We have added the reference (44) to the equation (1) and (2).
8. There are no references to reactions (3), (4) and (5) in the text. I guess that they are Fenton-like reactions.
The references to the reactions (3), (4) and (5) were listed in section reference (46, 47).
9. The first sentence of Chapter 3.3. should be written as follows: “INP can degrade many organic dyes through the following Fenton-like reactions (3)-(5) [46, 47]...”
We have changed it in the text.
10. Fig. 4.: What method was used to calculate the error bars presented in the figure?
Thank you very much for your suggestion, the value of the error bar is the difference between the two sets of repeated test data.
11. Fig. 5: The OX axis of Fig. 5d. should be described.
The OX axis of Fig. 5d presents a different kind of scavengers, which were given in Fig. 5d.
Reviewer 2 Report
This paper deals with the study of the degradation of Rh 6G with a montmorillonite/Fe NP under microwave treatment. The manuscript is quite well organized and reports some interesting information on the subject, however in my opinion, some aspects could be clarified.
In particular, the authors should take into consideration the following points:
General consideration: In my opinion is not totally correct called “irradiation” the use of microwave in this application. “microwave treatment” could be more appropriate.
1. Although the microwave treatment leads to obtain higher performance compared to other AOP process as showed in the figure 4f the authors have an idea to a possible application of this treatment for a practical point of view? The use of microwave can be ideal and easy for a laboratory scale but for the real wastewater treatment what is the possible solution to can use them?
2. Introduction: To facilitate the reading of the text please re-specify the abbreviation (as Inp and Mt).
-Lines 57-58: The authors should better explain the meaning of the phrase “…change the thermodynamic reduced..” I think that the microwave treatment modifies the reaction system providing more energy, but not changing the thermodynamic properties of the system. In fact, the high supplied energy allows to overcome the activation energy hedge (without modify it), helping also to the break of chemical bonds. Please revised and improve these concepts.
3. Materials and Method:
-Synthesis of Inp/Mt: What is the nominal concentrations of Fe chosen for the preparation of the Inp/Mt sample? What is the different respect to the value measured by XPS?
4. Result and discussion:
-Lines 101-102: The figure 1 should be better described and commented.
-Lines 108-109: The authors should improve the explanation of the XRD patterns. In my opinion the signal of iron nanoparticles is absent in the XRD pattern of Inp/Mt for the high and homogeneous dispersion of them in the Mt support. Please clarify this point.
5. XPS characterization: The XPS figure can be separated to Fig. 2 referring to totally different investigated properties. Moreover, the authors should provide the quantitative estimation of the iron detected in the surface of the sample by XPS. It could be useful to have an idea of the distribution of iron on the support.
-Line 221: Fig. 5: The authors should comment the performance obtained with AgNO3 and TEOA additives. Why with TEOA the degradation % is the same of not-treated system and on the contrary with AgNO3 the activity is lower?
For my point of view, the manuscript can be published in Materials after major revision according to the above reported comments.
Author Response
This paper deals with the study of the degradation of Rh 6G with a montmorillonite/Fe NP under microwave treatment. The manuscript is quite well organized and reports some interesting information on the subject, however in my opinion, some aspects could be clarified.
In particular, the authors should take into consideration the following points:
1. General consideration: In my opinion is not totally correct called “irradiation” the use of microwave in this application. “microwave treatment” could be more appropriate.
We have made corresponding changes in the text.
2. Although the microwave treatment leads to obtain higher performance compared to other AOP process as showed in the figure 4f the authors have an idea to a possible application of this treatment for a practical point of view? The use of microwave can be ideal and easy for a laboratory scale but for the real wastewater treatment what is the possible solution to can use them?
Thank you very much for your suggestion, the microwave is currently widely used in industry, such as food industry, medical, chemical industry, and also in wastewater treatment.
3. Introduction: To facilitate the reading of the text please re-specify the abbreviation (as Inp and Mt).
We have changed the abbreviation to nZVI and MMT.
4. Lines 57-58: The authors should better explain the meaning of the phrase “…change the thermodynamic reduced..” I think that the microwave treatment modifies the reaction system providing more energy, but not changing the thermodynamic properties of the system. In fact, the high supplied energy allows to overcome the activation energy hedge (without modify it), helping also to the break of chemical bonds. Please revised and improve these concepts.
We have changed it in section Induction and Reference.
- Materials and Method:
5. Synthesis of Inp/Mt: What is the nominal concentrations of Fe chosen for the preparation of the Inp/Mt sample? What is the different respect to the value measured by XPS?
Thank you very much for your suggestion. In section 2.2, 36.96 mmol Fe was dissolved in 50mL solution, so the concentrations of Fe is 0.7392 mol/L. Different binding energy represents different valence states of iron. The peaks at binding energies of 706.3 eV and 719.7 eV could be assigned to the 2p3/2 and 2p1/2 peaks of Fe0, respectively. The other two peaks at binding energies of 711.5 eV and 725.0 eV could be attributed to the 2p3/2 and 2p1/2 peaks of Fe (III), respectively, indicating that partial Fe0 on the surface had been oxidized to Fe (III).
-Result and discussion:
6. Lines 101-102: The figure 1 should be better described and commented.
We have made corresponding changes in the text.
7. Lines 108-109: The authors should improve the explanation of the XRD patterns. In my opinion the signal of iron nanoparticles is absent in the XRD pattern of Inp/Mt for the high and homogeneous dispersion of them in the Mt support. Please clarify this point.
Thanks a lot for your suggestion, it is really difficult to use XRD when the iron is highly dispersed on montmorillonite. However, the detection limit of XRD is also related to the iron content in the sample and the amount of sample used in the test. The iron content of the composite material is 15.35%, and the sample table area used by us during the test is also relatively large, which is equivalent to prolonging the scanning time, so iron can be detected from the composite material.
8. XPS characterization: The XPS figure can be separated to Fig. 2 referring to totally different investigated properties. Moreover, the authors should provide the quantitative estimation of the iron detected in the surface of the sample by XPS. It could be useful to have an idea of the distribution of iron on the support.
Thank you very much for your suggestion, although Fig. 2 referring to totally different investigated properties, they are all about discussing the antioxidant properties of composites, so they can be put together. Moreover, we have added a quantitative estimation of the iron in this section.
9. Line 221: Fig. 5: The authors should comment the performance obtained with AgNO3 and TEOA additives. Why with TEOA the degradation % is the same of not-treated system and on the contrary with AgNO3 the activity is lower?
TEOA is a typical hole-trapping agent and AgNO3 is a typical electron capture agent, indicating that holes are not a key component in the degradation of Rh 6G, and electrons have a weak effect on the degradation of Rh 6G.
For my point of view, the manuscript can be published in Materials after major revision according to the above reported comments.
Round 2
Reviewer 1 Report
My major concerns have been adequately addressed. The paper is now publishable.
Reviewer 2 Report
The manuscript is bow ready for the publication